# Developing Predictive Equations for Water Capturing Performance and Sediment Release Efficiency for Coanda Intakes Using Artificial Intelligence Methods

Oğuz Hazar [1], Gokmen Tayfur [1,*], Sebnem Elçi [1] and Vijay P. Singh [2]

1   Department of Civil Engineering, Izmir Institute of Technology, Izmir 35430, Turkey;
    oguzhazar@iyte.edu.tr (O.H.); sebnemelci@iyte.edu.tr (S.E.)
2   Department of Biological and Agricultural Engineering, Zachry Department of Civil and Environmental
    Engineering, Texas A&M University, Bizzell St, College Station, TX 78503-8879, USA;
    vijay.singh@agnet.tamu.edu
*   Correspondence: gokmentayfur@iyte.edu.tr

**Abstract:** Estimation of withdrawal water and filtered sediment amounts are important to obtain maximum efficiency from an intake structure. The purpose of this study is to develop empirical equations to predict Water Capturing Performance (WCP) and Sediment Release Efficiency (SRE) for Coanda type intakes. These equations were developed using 216 sets of experimental data. Intakes were tested under six different slopes, six screens, and three water discharges. In SRE experiments, sediment concentration was kept constant. Dimensionless parameters were first developed and then subjected to multicollinearity analysis. Then, nonlinear equations were proposed whose exponents and coefficients were obtained using the Genetic Algorithm method. The equations were calibrated and validated with 70 and 30% of the data, respectively. The validation results revealed that the empirical equations produced low MAE and RMSE and high $R^2$ values for both the WCP and the SRE. Results showed outperformance of the empirical equations against those of MNLR. Sensitivity analysis carried out by the ANNs revealed that the geometric parameters of the intake were comparably more sensitive than the flow characteristics.

**Keywords:** Coanda intake; dimensionless parameters; ANN; multicollinearity analysis; empirical equations; GA; MNLR; calibration; validation

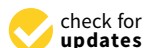



## 1. Introduction

Intake structures are used to divert water from channels and river systems for various purposes, such as energy production, irrigation, and domestic use [1–3]. Tyrolean and Coanda types of water intake structures are the most widely used bottom intake structures in the world. No matter the type of intake, the expected purpose from any intake structure is to supply required water while filtering most of sediments and other unwanted particles as much as possible [4,5]. This is because energy production stages are carried out with different types of high-value machinery developed for working under clear water conditions. They are sensitive to sediment particles within water. In addition, sediment particles can become a shelter for various types of bacteria and protozoa and reduce sanitation efficiency, especially for ultraviolet disinfection operations [6]. In addition, heavy metals can become attached to particles, contaminating water in time. Therefore, water that is not purified well from sediment and other particles can cause important health problems.

Withdrawal water and filtered sediment amounts depend on both structural design parameters such as bar spacing of an intake, screen length, screen slope inclination, etc. and incoming flow conditions such as discharge rate and sediment concentration of incoming flow. Therefore, estimation and determination of withdrawal water and excluded sediment amounts are highly important to obtain maximum efficiency from an intake structure.

Some researchers have tried to find an optimum design to overcome the clogging problem by performing experimental studies. A series of experiments were performed by Orth, et al. [7] using Tyrolean type water intakes. They have proposed that a bar profile with a rounded top has high sediment retention and clogging. Rounded shape bars were found to be more susceptible to clogging by Krochin and Sviatoslav [8], who have recommended that screen bars should be made of iron and their shapes should be rectangular or trapezoidal, bar spacing should range between 2 and 6 cm, and screen inclination should be 20. Bouvard [9] worked with Tyrolean type intakes and expressed that screen slope should be between 10 and 60% to avoid clogging. In the case of screening for hydroelectric power plant operations, bar spacing was expressed by Raudkiwi [10] as at least 5 mm and the screen slope as 20% to overcome any possible clogging problem.

There is a structural difference between Tyrolean and Coanda intakes. Tyrolean-type water intakes have straight screen bars which are oriented parallel to flow direction. On the other hand, Coanda-type water intakes have concave screen geometry where screen bars are placed perpendicular to the flow direction. An increment on screen slope reduces water column height on the intake screen, reducing both the orifice effect and withdrawn water discharge. Effect of screen inclination on withdrawn water for Tyrolean-type water intakes was studied by Castillo, et al. [11]. According to their clear water experiments, the best water capturing performance was obtained at 0% screen inclination and the worst results were obtained at 30%. On the other hand, in the case of sediment-laden flow, maximum water capture performance was obtained at 30% screen inclination and the worst result was obtained at 0% [11]. The difference is caused by screen clogging due to sediment particles. On the other hand, when Coanda intake is used instead of a Tyrolean intake, even in steeper screen inclinations, the shear mechanism becomes more dominant, keeping withdrawal water discharge relatively high. The self-cleaning ability and high-water withdrawal capacity make Coanda-type water intakes more preferable than Tyrolean ones. In addition, the study of Nøvik, et al. [12] indicates that Coanda screens mostly show satisfactory performance under cold climate conditions. Furthermore, Coanda type water intakes are environmentally friendly structures since they can allow fish and invertebrates to pass through downstream of a river [13]. Hence, this study has focused on Coanda-type water intakes.

An important study on Coanda intake structures was done by Wahl [14], who has proposed empirical equations for offset height of screen bar and orifice effect. He mentions that wire tilt angle, which is directly related to offset height, affects the screen capacity. It increases the shear effect and withdrawal of water quantity. However, it can cause some disadvantages to the screen performance such as sediment retention and clogging of the screen. He also concludes that changing sloth width or wire size directly affects the screen porosity and the screen flow capacity. In another study, Wahl [15] investigated the effect of changes in screen parameters, such as wire tilt angle, screen curvature (arc) radius, and screen length on the withdrawal water discharge. The studies of Wahl [14,15] are important for investigating the effect of various screen parameters on the unit withdrawal discharge under clear water conditions. A numerical model for clear water conditions to predict water discharge through the intake in case of different screen design parameters and variations was developed by Dzafo and Dzaferovic [16]. Another numeric model to analyze flow in a diversion channel in order to indicate how the numerical (Delft3D-FLOW) and physical models can be used to observe flow patterns nearby a diversion channel, with Coanda intake to estimate design parameters, was developed by Hosseini and Coonrod [17].

In real-life applications, Coanda type intake structures face sediment-laden flow conditions as the other intake structures. Some studies have considered sediment-laden flow for Coanda type intakes. For example, a series of experimental studies were performed by Howarth [18] to investigate Coanda screens. Three Coanda screens that have different sloth widths (bar opening) were used by Huber [19] who has indicated that the sediment exclusion efficiency is increasing with decreasing sloth width. On the other hand, a smaller sloth width increases the risk of clogging. Some experiments were performed by May [20]

by considering both clear water and sediment-laden flow using three different Coanda screens which had different screen openings. May [20] summarizes that screens having smaller wire openings show good performance for sediment exclusion but are more susceptible to clogging. Both studies [19,20] have investigated the effect of different discharge rates and sloth widths. However, constant screen slope and curvature radius were used in their studies. On the other hand, a series of experiments at Izmir Institute of Technology (IZTECH) Hydraulic Laboratory was performed by Hazar and Elci [21] by using sediment-laden flows using Coanda intakes. Parameters of Water Capturing Performance (WCP) and Sediment Release Efficiency (SRE) were defined to explain the screen performances under different conditions. The multi-linear equations for both WCP and SRE of Coanda screens were developed using the linear regression as a statistical analysis method. However, these equations were not validated since all the data were employed in their construction.

The relations between WCP and related parameters of water flow, sediment, and intake characteristics are not linear but rather highly nonlinear, which is also true for SRE. This implies that nonlinear empirical equations can represent the actual physical processes. The advantage of developing such empirical equations can be beneficial for designing optimal intakes and for predicting diverted water amount and corresponding sediment concentration in the diverted water. Hence, there is a need for developing nonlinear empirical equations to predict WCP and SRE as a function of Coanda intake structural characteristics, water, fluid, and sediment parameters. This study would be the first one, to the knowledge of the authors, in the literature to develop such empirical equations. To develop the equations, the dimensionless parameters are first to be subjected to the multicollinearity analysis. Empirical nonlinear equations, whose coefficients and exponents would be determined by applying the method of Genetic Algorithm (GA), would be proposed. The construction of the empirical equations would be carried out by using 70% of the data for the calibration and 30% for the validation.

## 2. Methods and Methodology

### 2.1. Experimental Setup and Data

The experimental set-up and results were already presented by Hazar and Elci [21]. Experiments were performed at IZTECH Hydraulics Laboratory in Izmir, Turkey (Figure 1). Six different Coanda screens having different properties were designed (Table 1). Each Coanda screen had total screen length of 100 cm, net screen length of 60 cm, and screen width of 40 cm.

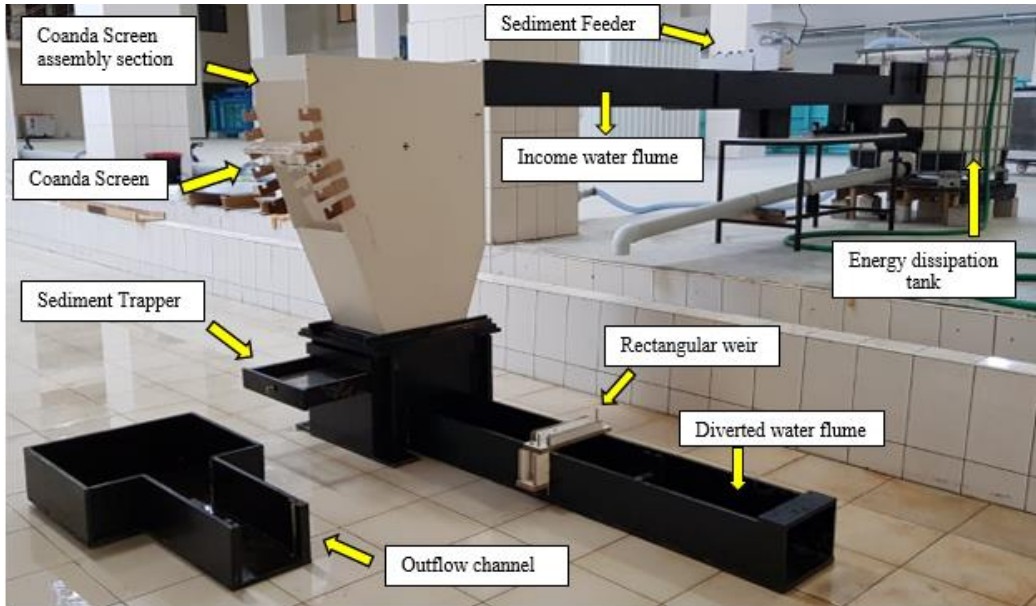

**Figure 1.** Experimental setup utilizing different Coanda-type intakes in the experiments.

**Table 1.** Screen characteristics.

| Screen Type | Sloth Width (mm) | Curvature Radius (mm) | Void Ratio (e/a) |
|---|---|---|---|
| Coanda R800 (1) | 1 | 800 | 0.046 |
| Coanda R800 (2) | 2 | 800 | 0.092 |
| Coanda R800 (3) | 3 | 800 | 0.138 |
| Coanda R1200 (1) | 1 | 1200 | 0.046 |
| Coanda R1200 (2) | 2 | 1200 | 0.092 |
| Coanda R1600 (1) | 1 | 1600 | 0.046 |

To observe the screen inclination effect on the screen performances, wooden sockets which allow the user to adjust a screen inclination in the range of 0 and 30 degrees (0, 5, 15, 20, 25, 30 degrees) were mounted on the intake body section walls (Figure 2). During the experiments, three different water discharges, 2.4, 5.56, and 7.96 L/s, were used to observe the incoming flow effect on both WCP and SRE.

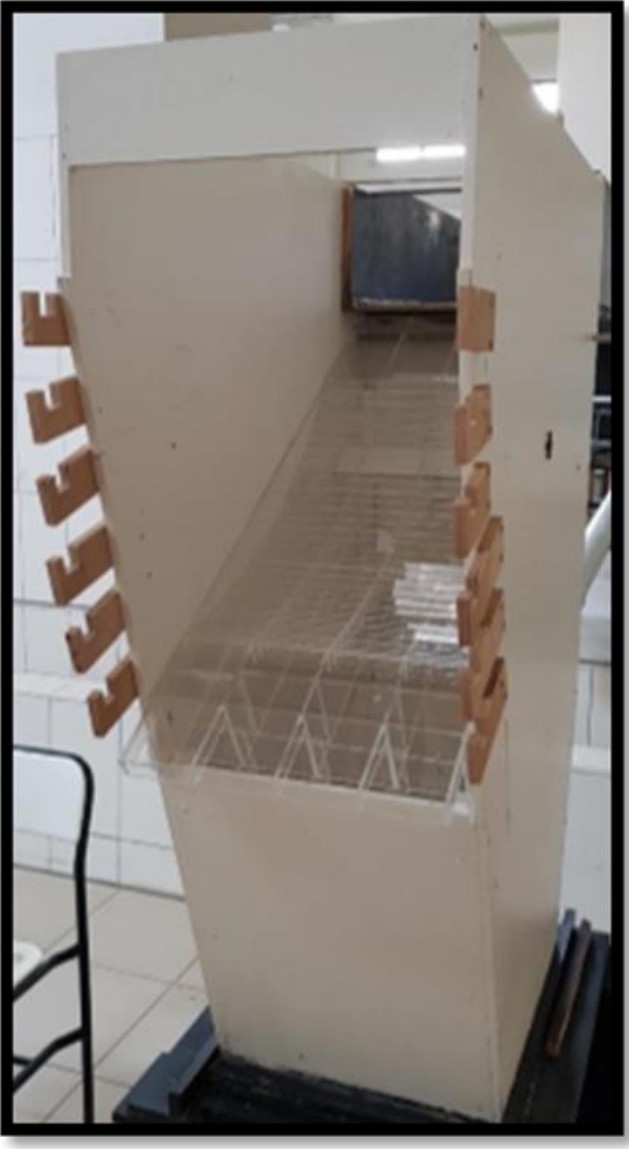

**Figure 2.** Wooden sockets used to adjust slope of the Coanda screens.

In the case of sediment-laden flow, a novel sediment feeder structure was designed to allow a user to adjust sediment concentration during the experiments (Figure 3). Uniform sediment particles that had 0.8 mm diameter were used with 300, 695, and 995 g amounts for 2.4, 5.56, and 7.96 L/s, respectively, to obtain the same sediment concentration of 125 g/L for each discharge case.

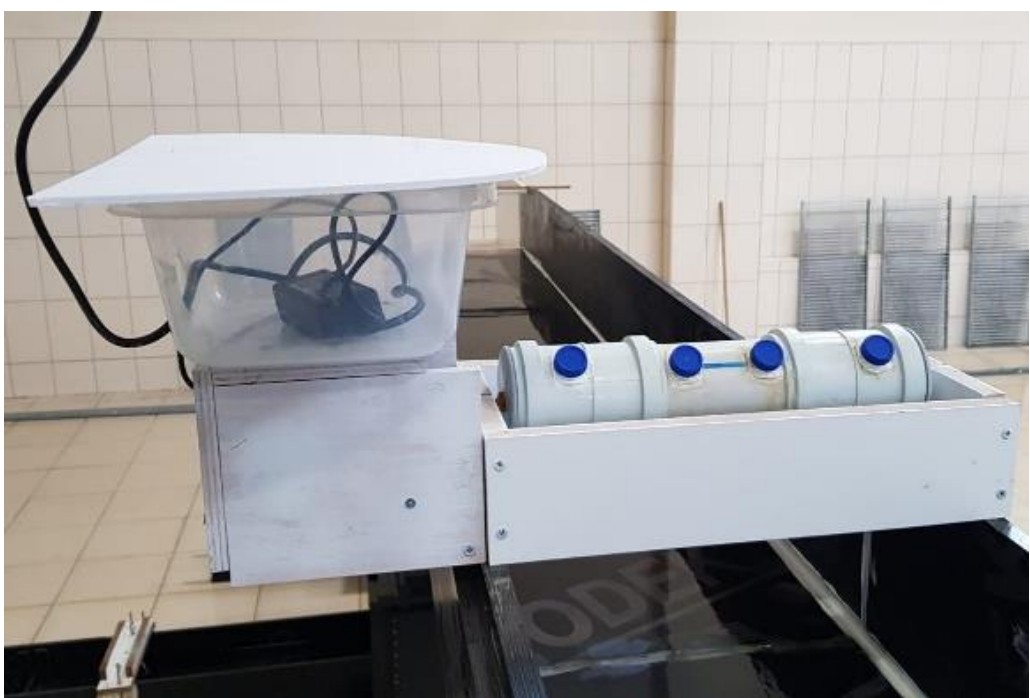

**Figure 3.** Sediment feeder device.

There were 108 cases (6 different slopes × 6 different screens × 3 different flow discharge) and each experiment was repeated three times, and an average value of WCP was obtained for each experiment. Thus, in total, 108 × 3 = 324 experiments were carried out and 108 average WCP values were used in the analysis. The similar procedure was also applied to the SRE and 108 average SRE values were used in the analysis. Note that the experiments for the WCP were done in clear water while the experiments for the SRE were carried out in sediment-laden flows and thus they were totally different experiments. The statistics of the measured WCP and SRE from all the experiments are summarized in Table 2. Details of the experimental setup and the experiments can be obtained from Hazar and Elci [21].

**Table 2.** Statistical summary of data sets for WCP and SRE.

| Data Sets | WCP ($Q_{in}/Q_{div}$) | SRE ($S_{in}/S_{re}$) |
|---|---|---|
| Maximum | 100 | 90.4 |
| Minimum | 38.7 | 0.3 |
| Range | 61.3 | 90.0 |
| Mean | 70.3 | 52.7 |
| St. Deviation | 16.8 | 26.0 |

For developing the empirical equations, dimensionless parameters were developed, as presented in Table 3.

**Table 3.** Dimensionless parameters.

| Dimensionless Parameters | Description |
|---|---|
| $\theta$ | Screen Slope (degree) |
| L/R | Net screen length/Screen curvature radius |
| m = e/a | Bar openings area/Total screen area |
| e/R | Bar opening/Curvature radius |
| $Fr_{(R)} = \dfrac{V}{\sqrt{gR}}$ | Froude number based on screen curvature radius |
| $We_{(R)} = \dfrac{\rho V^2 R}{\sigma}$ | Weber number based on screen curvature radius |
| $D_{50}/R$ | Median of the sediment diameter/Flow depth at beginning of the screen |
| $D_{50}/e$ | Median of the sediment diameter/Bar opening |
| WCP $(Q_{in}/Q_{div})$ | Water capturing performance |
| SRE $(S_{in}/S_{re})$ | Sediment release efficiency |

Definition of parameters: $\theta$ is the screen slope, L is the net screen length, R is the screen curvature radius, a is the distance between mid-points of two consecutive screen bars, e is the net opening between two consecutive bars, V is the water velocity, g is the gravitational acceleration, $\rho$ is the water density, $\sigma$ is the surface tension, d50 median of the sediment diameter, Qin is the incoming discharge through the intake, Qdiv is the diverted water discharge by the intake, Sin is the total sediment amount which was fed into the main flume, and Sre is the released sediment amount by the intake.

*2.2. Multicollinearity Analysis*

The main aim of this study is to develop nonlinear empirical equations for WCP and SRE as functions of fluid, sediment, flow, and intake structure parameters for Coanda intakes. Evidently, there are 14 parameters (see Table 3), and considering all of them in any equation could be cumbersome work, which may result in a non-practical equation. Therefore, as a first step, the number of parameters was reduced by creating the dimensionless parameters, as presented in Table 3, where 10 of them are defined. The objective is to develop an empirical equation that can be as comprehensive (reflecting the actual physical process) as possible but at the same time simple and user-friendly. Before constructing the empirical equations, the multicollinearity analysis was performed to overcome any possible collinearity problem.

Multicollinearity refers to one predictor or independent variable in a regression model being able to be linearly predicted from other independent variables. It can cause a high variance of the estimated coefficients, not being able to reflect correct values. The Variance Inflation Factor (VIF) is one of the common methods that is used to distinguish collinearity in an analysis [22]. A small VIF value indicates a low correlation among predictor variables. There exist some suggestions for acceptable VIF value in the literature. Generally, a value of 10 is suggested as a top limit; VIF $\geq$ 10 indicates high multicollinearity among parameters. Thus, it is acceptable if the VIF is less than 10 [23]. The VIF is defined as follows [24]:

$$VIF = \frac{1}{1 - R^2} \tag{1}$$

where R is the coefficient of determination.

Six parameters ($\theta$, m, e/R, L/R, Froude, and Weber) for the WCP and 8 parameters ($\theta$, m, e/R, L/R, Froude, Weber, D50/R, and D50/e) for the SRE were subjected to the multicollinearity analysis. The parameters having acceptable VIF values are presented in Table 4 for the WCP and in Table 5 for the SRE.

**Table 4.** Independent parameters and VIF values for WCP.

| Independent Parameters | VIF Value |
|---|---|
| θ (Screen Slope) | 1.00 |
| m = (e/a) | 1.18 |
| L/R | 6.67 |
| Froude | 5.56 |
| Weber | 6.25 |

**Table 5.** Independent parameters and VIF values for SRE.

| Independent Parameters | VIF Value |
|---|---|
| θ (Screen Slope) | 1.00 |
| L/R | 5.88 |
| Froude | 5.88 |
| Weber | 6.25 |
| D50/e | 1.11 |

*2.3. Genetic Algorithm (GA)*

Genetic Algorithm is a nonlinear search and optimization method, inspired by biological processes of natural selection and survival of the fittest. GA has two main units as gene and chromosome. A gene consists of bits and a chromosome consists of genes. The gene represents a model parameter to be optimized and each chromosome stands for a solution candidate. In GA, the search process is initiated with many chromosomes. In each iteration, search space is scanned by the chromosomes, while the fitness evaluation, selection, pairing, crossover, and mutation operations are performed. Figure 4 shows the flowchart summarizing the GA operations. There are innumerous studies (papers and books) available in the literature, including [25], on the details of the GAs and their operations, and GA applications in water resources engineering.

---

**Forming initial gene pool**

(Randomly generating chromosomes)

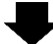

**Evaluating fitness of each chromosome**

(Determining fitness value by inserting each chromosome to objective function)

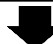

**Selection**

(Selecting fit chromosomes and eliminating weak ones)

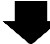

**Cross-over**

(Exchanging genes of parent chromosomes to generate new offspring)

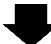

**Mutation**

(Generating newer offspring and preventing getting trapped into local minimum)

---

**Figure 4.** Flow chart for Genetic Algorithm operations.

## 2.4. Artificial Neural Networks (ANNs)

Artificial Neural Networks were employed in this study to compare the prediction performance against the empirical equations. ANNs have been developed as analogies to the human brain system, consisting of many artificial neurons, with connection links and layers that, as a whole system, can learn from experience and experiments and store information.

Figure 5 shows a typical commonly employed single hidden layer network where inputs ($X_i$) are passed on to neurons at the hidden layer as $X_iV_{ij}$. These neurons, in turn, sum the weighted input as $net_j = \sum_{i=1}^{n}(X_iV_{ij})$ which is then transferred by a nonlinear transfer function (mostly, the sigmoid or the tangent hyperbolic function) to produce an output. This response can become an input for other neurons located in the next layer. This is continued until the network produces an output ($Z_{model}$). The propagation from input to hidden to output neurons is called a forward pass.

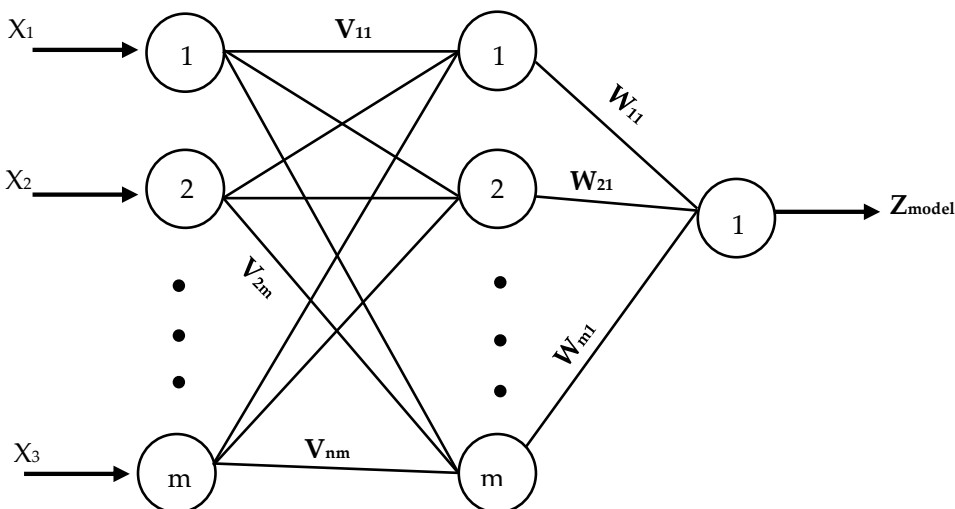

**Figure 5.** Schematic representation of a single hidden layer ANN.

The back propagation algorithm is generally employed to train the network by finding optimal values of the connection weights ($V_{ij}$, $W_{ij}$) that can produce an output vector $Z = (z_1, z_2,..., z_p)$ as near as possible to target output vector $T = (t_1, t_2,...,t_p)$. For each input pattern (p), the network produces an output and the related error is computed and then the total error ($E_{total}$) is obtained by summing each error as follows [25]:

$$E_{total} = \frac{1}{2}\sum_{p=1}^{N}(z_p - t_p)^2 \qquad (2)$$

where $z_p$ is the model produced output for p-pattern, $t_p$ is the target (actual) output for p-pattern, and N is the number of training patterns.

The total error is then back propagated from output neuron to inner to input neurons by tuning connection weights (Figure 5). The error propagation from output neuron to hidden layer to input layer is called the backward pass. A single forward and backward pass constitutes a single iteration. At each iteration, the connection weights are updated (optimized) as follows [25], by minimizing the total error ($E_{total}$):

$$v_{ij}^{new} = v_{ij}^{old} - \delta\frac{\partial E_{total}}{\partial v_{ij}} \qquad (3)$$

where $v_{ij}^{new}$ and $v_{ij}^{old}$ are current and previous values of connection weights at successive iterations. $\delta$ is the learning rate which can assume values greater than zero and less than one.

As the iterations are continued, the total error is expected to decrease. There are innumerous studies (papers and books) available in the literature, including, [25], on the details of the feed forward networks, the back propagation algorithm, the network training and testing, and ANN applications in water resources engineering.

### 2.5. Constructing Empirical Equations

This study proposes nonlinear empirical equations for predicting WCP and SRE, based on the related dimensionless parameters presented in Tables 4 and 5. As presented above, e/R is excluded from the WCP model after the multicollinearity analysis (see Table 4). Thus, the remaining parameters ($\theta$, m, L/R, Froude, and Weber) are considered in the empirical equation for the WCP. However, another important issue is to decide the form of the equation, i.e., how to place each parameter (as numerator or denominator) with a positive exponent in the equation. To do so, the behavior (direct or inverse variation) between each parameter and the output variable, WCP, is analyzed. According to Figure 6, WCP varies directly with the void ratio (m) while it varies inversely with Weber number, Froude number, and the screen slope ($\theta$). There is no clear relation observed between L/R and WCP. Therefore, the following nonlinear equation is proposed for WCP:

$$\text{WCP} = c_1 \left[ \frac{(m)^{a_2} \left( \frac{L}{R} \right)^{a_5}}{(\theta)^{a_1} (\text{Fr})^{a_3} (\text{We})^{a_4}} \right] \tag{4}$$

where $c_1$ is a coefficient and $a_1$, $a_2$, $a_3$, $a_4$, and $a_5$ are exponents whose optimal values are obtained by the GA. The optimal values of $c_1$, $a_1$, $a_2$, $a_3$, and $a_4$ are searched within the positive range while the search space covered a wide range (positive to negative) for $a_5$ since there is no clear variation pattern between L/R and WCP, as more details are given in the next section.

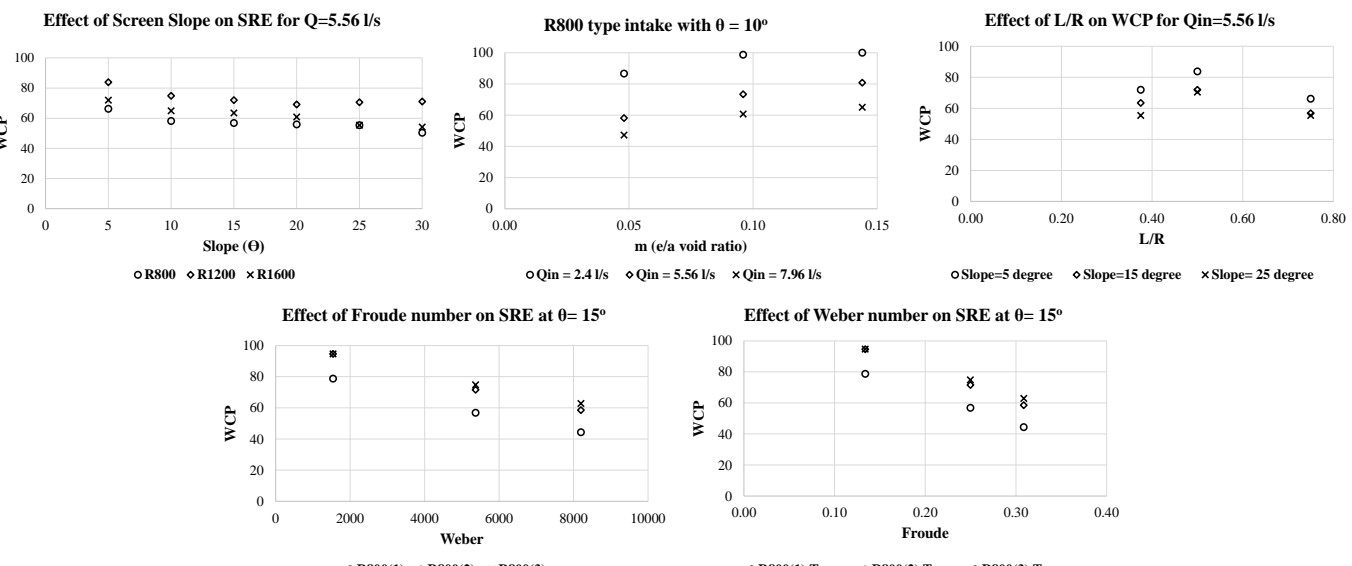

**Figure 6.** Variation of $\theta$, m, L/R, Froude, and Weber number parameters with WCP.

As presented above, m, D50/R, and e/R are excluded from the SRE empirical model after the multicollinearity analysis and therefore the remaining parameters, presented in Table 5 ($\theta$, D50/e, Froude, Weber, and L/R), are considered in the empirical equation for the SRE. To decide the form of the equation, as it is presented above for the WCP, the behavior (direct or inverse variation) between each parameter and the output variable, SRE, is first analyzed. According to Figure 7, SRE varies directly with screen slope ($\theta$), Froude,

and Weber. There is no clear variation behavior observed for L/R and D50/e. Hence, the following nonlinear empirical equation is proposed for the SRE:

$$\text{SRE} = c_2 \left[ \frac{(\theta)^{b_1} (\text{Fr})^{b_2} (\text{We})^{b_3}}{\left(\frac{\text{D50}}{\text{e}}\right)^{b_4} \left(\frac{\text{L}}{\text{R}}\right)^{b_5}} \right] \tag{5}$$

where $c_2$ is a coefficient and $b_1$, $b_2$, $b_3$, $b_4$, and $b_5$ are exponents. The optimal values of $c_2$, $b_1$, $b_2$, and $b_3$ are obtained by the GA, searching values within the positive range, while the search space covered a wide range (positive to negative) for $b_4$ and $b_5$ since there is no clear variation pattern between these parameters and SRE. More details are given in the next section.

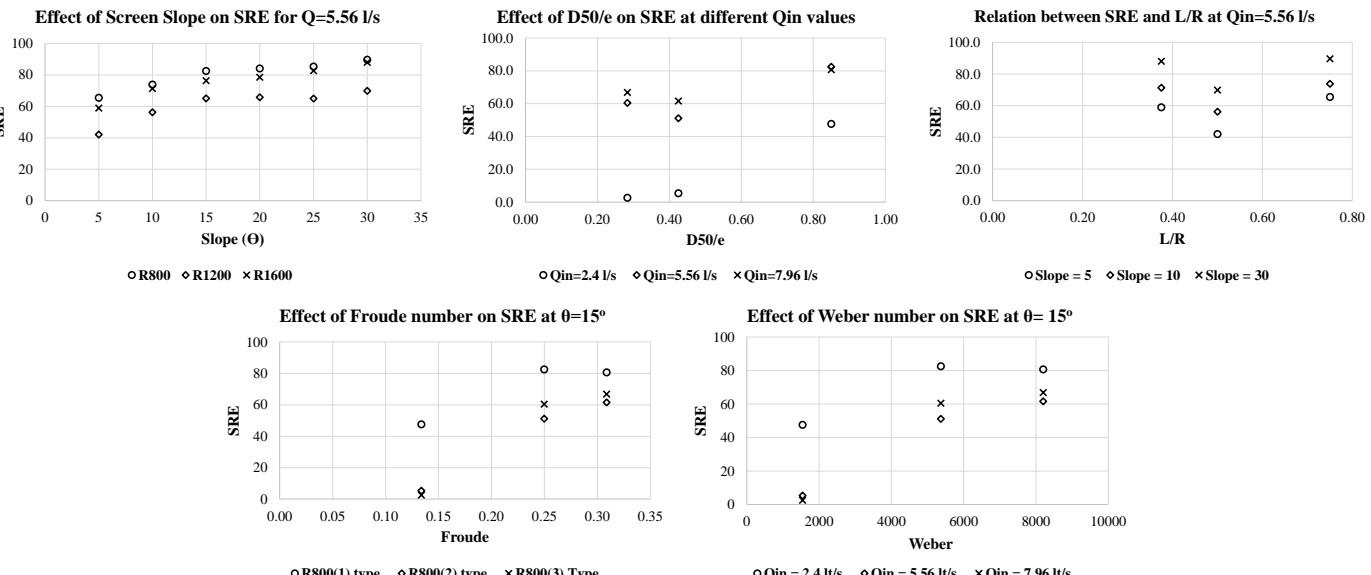

**Figure 7.** Variation of e/R, Weber, m, Slope, L/R, and D50/R parameters with SRE.

## 3. Results

### 3.1. GA-Based Empirical Equations

Optimal values of coefficients and exponents of Equations (4) and (5) were found by employing the GA. For this purpose, a total of 216 data sets were gathered from the experimental study: 108 for the WCP and 108 for the SRE. Of the 108 sets for WCP, 70 were randomly selected for calibration, while the rest were for validation. The same was done for SRE. The mean absolute error functions were minimized while finding the optimal values of the coefficients and exponents of Equations (4) and (5). The respected error functions can be expressed as follows:

$$\text{MAE} = \frac{1}{N} \sum_{i=1}^{N} \text{abs}(\text{WCP}_{\text{model}} - \text{WCP}_{\text{measured}}) \tag{6}$$

$$\text{MAE} = \frac{1}{N} \sum_{i=1}^{N} \text{abs}(\text{SRE}_{\text{model}} - \text{SRE}_{\text{measured}}) \tag{7}$$

During iterations, 80% crossover rate, 4% mutation rate, and 6000 epochs were employed. The search space for $c_1$ and $c_2$ was set to 1–200 and for the exponents of $a_5$, $b_4$, and $b_5$ the search space was set to $-3$ and $+3$ while it was set to 0–3 for the other exponents. The obtained optimal values of the coefficients are shown in Table 6. Figures 8 and 9 present the validation and calibration stages for WCP and SRE, respectively. As seen, predictions are satisfactory for both cases for which the related error measures are summarized in Table 7.

**Table 6.** Optimal values of the coefficients for GA based equations.

| Parameter | $c_1$ | $c_2$ | $a_1$ | $a_2$ | $a_3$ | $a_4$ | $a_5$ | $b_1$ | $b_2$ | $b_3$ | $b_4$ | $b_5$ |
|---|---|---|---|---|---|---|---|---|---|---|---|---|
| WCP | 151.8 | - | 0.12 | 0.15 | 0.39 | 0.07 | 0.15 | - | - | - | - | - |
| SRE | - | 56.5 | - | - | - | - | - | 0.26 | 0.81 | 0.04 | −0.21 | 0.60 |

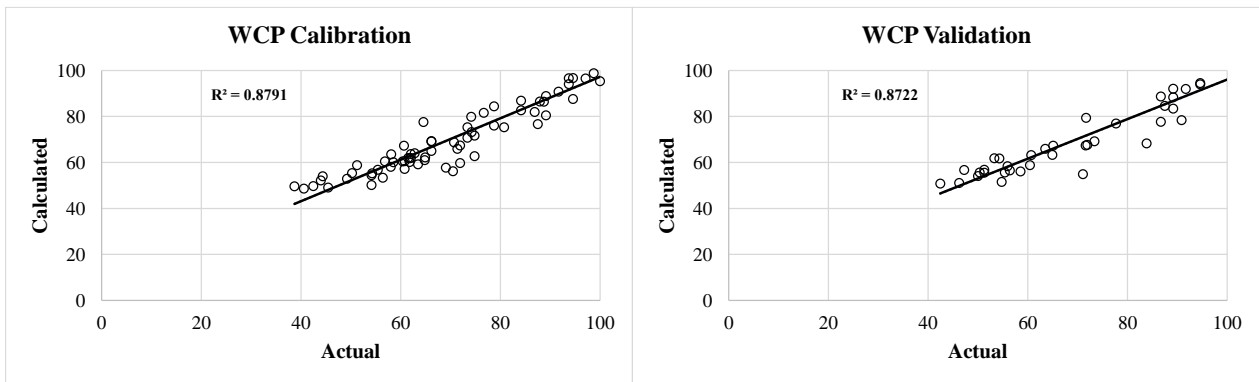

**Figure 8.** Calibration and validation stages for WCP.

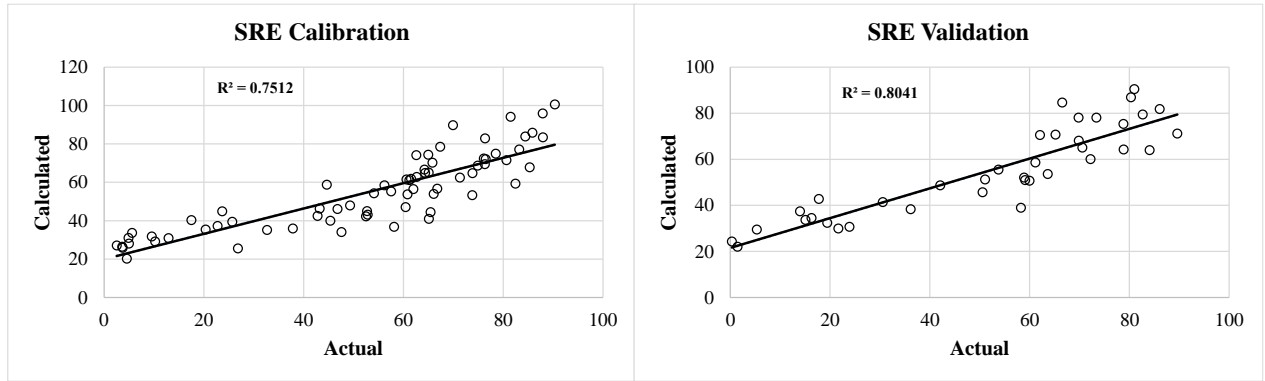

**Figure 9.** Calibration and validation stages for SRE.

**Table 7.** Error measures for WCP and SRE.

| GA | Calibration Stage | | Validation Stage | |
|---|---|---|---|---|
| | WCP | SRE | WCP | SRE |
| MAE | 4.32 | 10.32 | 4.71 | 10.77 |
| RMSE | 5.72 | 13.18 | 6.15 | 12.99 |
| $R^2$ | 0.88 | 0.75 | 0.87 | 0.80 |

The resulting empirical equations can be presented as follows:

$$\text{WCP} = 151.8 \left[ \frac{(m)^{0.15} \left(\frac{L}{R}\right)^{0.15}}{(\theta)^{0.12} (\text{Fr})^{0.39} (\text{We})^{0.07}} \right] \tag{8}$$

$$\text{SRE} = 56.5 \left[ \frac{(\theta)^{0.26} (\text{Fr})^{0.81} (\text{We})^{0.04} \left(\frac{D_{50}}{e}\right)^{0.21}}{\left(\frac{L}{R}\right)^{0.60}} \right] \tag{9}$$

### 3.2. MNLR Based Empirical Equations

The performances of the empirical equations were compared against those of the Multi Nonlinear Equations (MNLR). The proposed MNLR are expressed as follows:

$$WCP = c_1(\theta)^{a_1} + c_2(m)^{a_2} + c_3(Fr)^{a_3} + c_4(We)^{a_4} + c_5\left(\frac{L}{R}\right)^{a_5} \tag{10}$$

$$SRE = d_1(\theta)^{b_1} + d_2(Fr)^{b_2} + d_3(We)^{b_3} + d_4\left(\frac{D50}{e}\right)^{b_4} + d_5\left(\frac{L}{R}\right)^{b_5} \tag{11}$$

The same data used for the calibration of Equations (8) and (9) were employed for finding the coefficients and exponents of Equations (10) and (11). The obtained optimal values of the coefficients are presented in Tables 8 and 9. Figures 10 and 11 present the calibration and validation stages for WCP and SRE, respectively. The error measures related to these figures are summarized in Table 10.

**Table 8.** Optimal values of the coefficients for WCP in the case of MNLR equations.

| Parameter | $c_1$ | $c_2$ | $c_3$ | $c_4$ | $c_5$ | $a_1$ | $a_2$ | $a_3$ | $a_4$ | $a_5$ |
|-----------|-------|-------|-------|-------|-------|-------|-------|-------|-------|-------|
| WCP | 140.09 | 10.39 | 7.54 | 215.73 | 19.96 | −30.15 | 7.83 | −0.77 | −0.21 | 2.47 |

**Table 9.** Optimal values of the coefficients for SRE in the case of MNLR equations.

| Parameter | $d_1$ | $d_2$ | $d_3$ | $d_4$ | $d_5$ | $b_1$ | $b_2$ | $b_3$ | $b_4$ | $b_5$ |
|-----------|-------|-------|-------|-------|-------|-------|-------|-------|-------|-------|
| SRE | 1.83 | 21.00 | −217.90 | 51.67 | 0.82 | 0.85 | 20.00 | −9.81 | 0.47 | 0.43 |

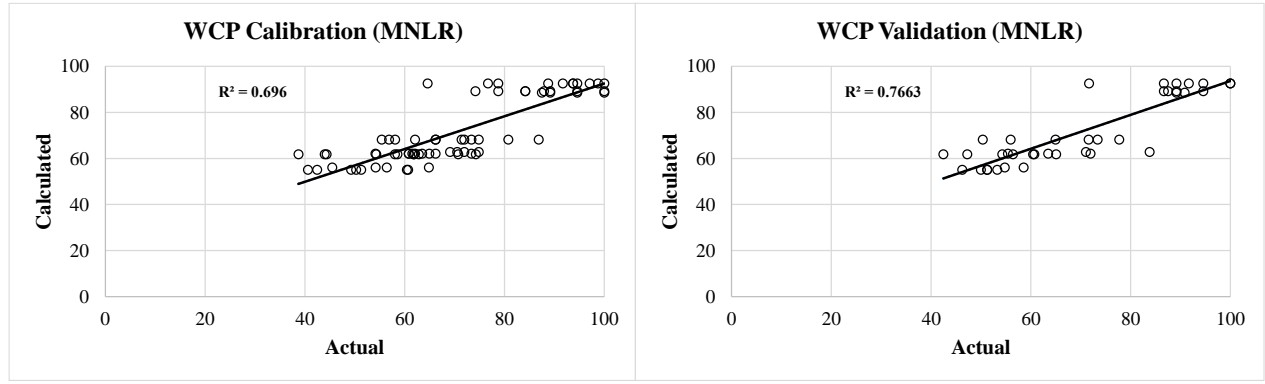

**Figure 10.** Calibration and validation stages for WCP (MNLR model).

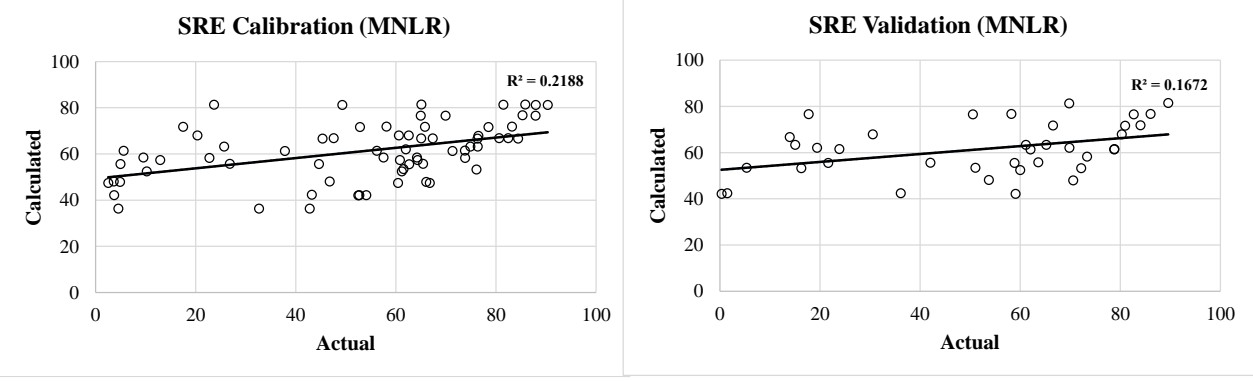

**Figure 11.** Calibration and validation stages for SRE (MNLR model).

**Table 10.** Error measures for WCP and SRE.

| MNLR | Calibration Stage | | Validation Stage | |
|---|---|---|---|---|
| | WCP | SRE | WCP | SRE |
| MAE | 6.95 | 18.04 | 6.28 | 20.14 |
| RMSE | 9.12 | 24.00 | 8.48 | 25.98 |
| $R^2$ | 0.70 | 0.22 | 0.77 | 0.17 |

The obtained MLNR equations can be expressed as follows:

$$WCP = 140.09(\theta)^{-30.15} + 10.39(m)^{7.83} + 7.54(Fr)^{-0.77} + 215.73(We)^{-0.21} + 19.96\left(\frac{L}{R}\right)^{2.47} \tag{12}$$

$$SRE = 1.83(\theta)^{0.85} + 21(Fr)^{20} - 217.9(We)^{-9.81} + 51.67\left(\frac{D50}{e}\right)^{0.47} + 0.82\left(\frac{L}{R}\right)^{0.43} \tag{13}$$

### 3.3. ANN Predictions

The ANN for the WCP has 5 neurons (input parameters: $\theta$ (angle), m (void ratio), L/R, Froude number, and Weber number) in the input layer, 11 neurons in the hidden layer, and a single output neuron for WCP (see Figure 12). The ANN of SRE contains 5 neurons (Input parameters: $\theta$ (angle), L/R, Froude number, Weber number, D50/e) in the input layer, 11 neurons in the hidden layer, and 1 output neuron for the SRE (see Figure 13). The networks were trained with 6000 epochs. The same data sets, which were randomly separated as about 70% for the calibration and 30% for the validation for the GA-based and MNLR empirical equations, were employed for the ANN training and testing stages, respectively.

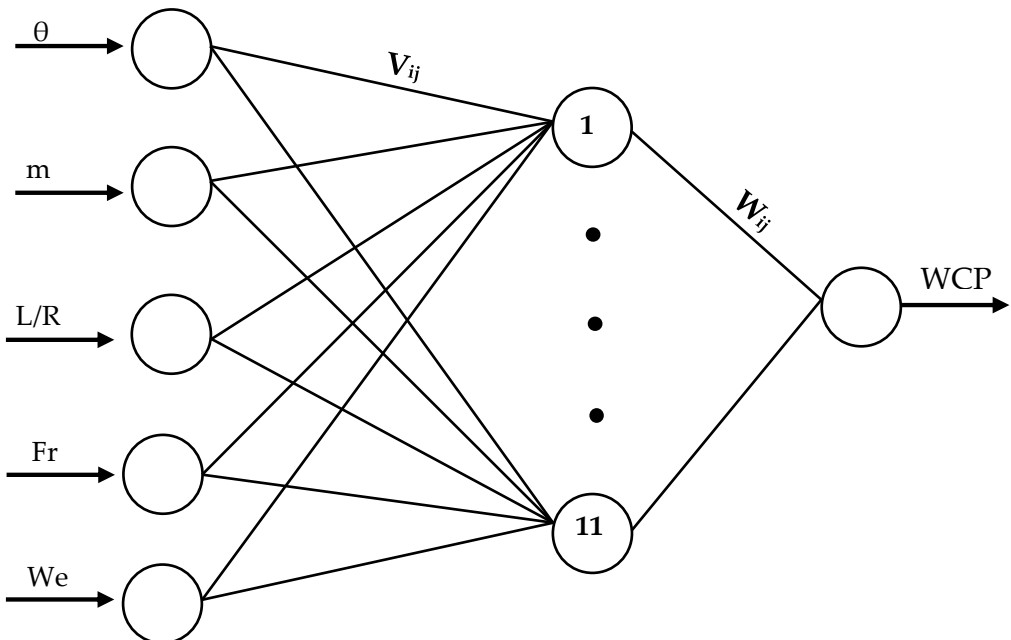

**Figure 12.** ANN configuration for WCP.

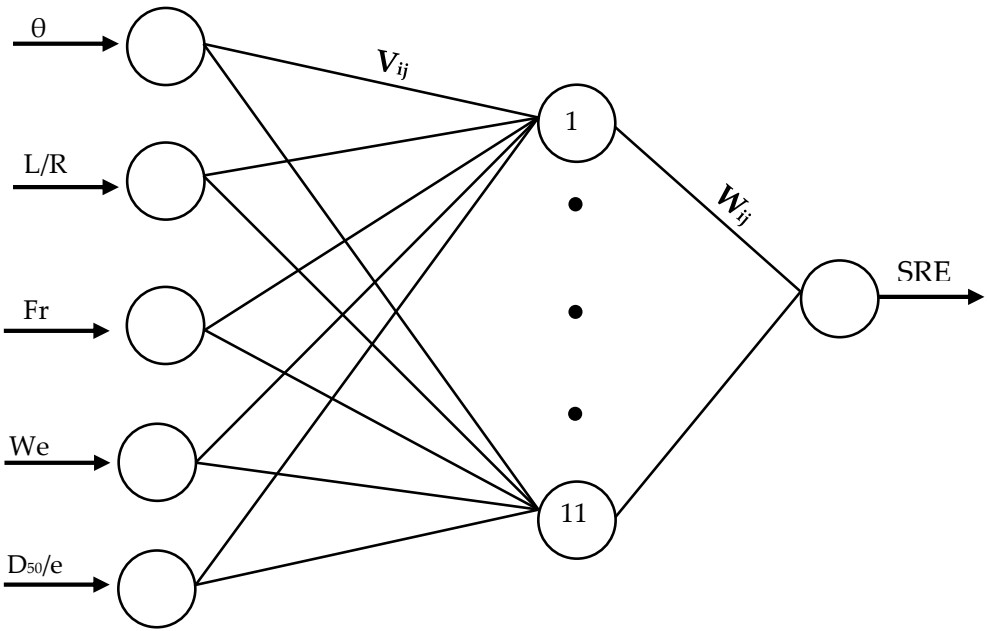

**Figure 13.** ANN configuration for SRE.

Table 11 presents the error measures for the WCP and SRE at the training and testing stages. As seen, for both the cases, ANNs produced low MAE and RMSE and high $R^2$ values.

**Table 11.** Error measures for WCP and SRE.

| ANN | Training Stage | | Testing Stage | |
|---|---|---|---|---|
| | WCP | SRE | WCP | SRE |
| MAE | 0.75 | 1.47 | 3.39 | 4.32 |
| RMSE | 1.08 | 2.37 | 4.30 | 5.37 |
| $R^2$ | 0.99 | 0.99 | 0.94 | 0.96 |

*3.4. Sensitivity Analysis by ANNs*

Sensitivity analysis was carried out by employing the ANNs to identify the sensitive parameters. Tables 12 and 13 summarize the results for WCP and SRE, respectively. As seen in Table 12, the geometric characteristics of θ (screen slope) and m = (ratio of bar openings area to total screen area) parameters turned out to be the most sensitive ones for the WCP. For the SRE, θ (screen slope) and D50/e (ratio of median of sediment diameter to bar opening) are the most sensitive ones (Table 13). It turns out that flow characteristics are not, comparatively, as sensitive as the intake geometric ones for the WCP and intake geometric and sediment characteristics for the SRE.

**Table 12.** Sensitivity analysis for WCP.

| Independent Parameters | MAE | RMSE | $R^2$ |
|---|---|---|---|
| θ (Screen Slope) | 4.259 | 5.475 | 0.892 |
| M = (e/a) | 4.670 | 6.132 | 0.865 |
| L/R | 1.220 | 1.826 | 0.988 |
| Froude | 1.353 | 1.886 | 0.987 |
| Weber | 1.209 | 1.666 | 0.990 |

**Table 13.** Sensitivity analysis for SRE.

| Independent Parameters | MAE | RMSE | $R^2$ |
|---|---|---|---|
| θ (Screen Slope) | 7.408 | 9.304 | 0.871 |
| L/R | 2.857 | 3.464 | 0.982 |
| Froude | 2.511 | 3.249 | 0.984 |
| Weber | 2.134 | 2.853 | 0.988 |
| D50/e | 9.033 | 11.24 | 0.812 |

## 4. Discussion

The calibration and validation of Equation (8), the GA-based empirical equation predicting WCP as a function of the intake geometric and flow characteristics variables, was successfully accomplished with low errors (MAE = 4.32, RMSE = 5.72 for the calibration stage, and MAE = 4.71, RMSE = 6.15 for the validation stage) and high $R^2$ values of 0.88 (calibration) and 0.87 (validation). When its success is compared against that of the MNLR equation (Equation (12)), it is clearly seen that the empirical equation outperformed the MNLR one, which produces relatively low $R^2$ = 0.70 and high errors of MAE = 6.95, RMSE = 9.12, at the calibration stage and MAE = 6.28, RMSE = 8.48, $R^2$ = 0.77 at the validation stage (see Table 14). The GA-based empirical model produced comparable results against those of the ANN, which is a very powerful soft computing method for nonlinear problems. Although ANN produced less errors and high $R^2$ values, as seen in Table 14, they do not yield any mathematical equation, as opposed to the empirical one.

**Table 14.** Performance comparison between GA_based empirical equation, MNLR, and ANN for WCP.

| WCP | Calibration Stage | | | Validation Stage | | |
|---|---|---|---|---|---|---|
| | GA | MNLR | ANN | GA | MNLR | ANN |
| MAE | 4.32 | 6.95 | 0.75 | 4.71 | 6.28 | 3.39 |
| RMSE | 5.72 | 9.12 | 1.08 | 6.15 | 8.48 | 4.30 |
| $R^2$ | 0.88 | 0.70 | 0.99 | 0.87 | 0.77 | 0.94 |

The calibration and validation of Equation (9), the GA-based empirical equation predicting SRE as a function of the intake geometric, fluid, sediment, and flow characteristics variables, was successfully accomplished with low errors (MAE = 10.32, RMSE = 13.18 for calibration, and MAE = 10.77, RMSE = 12.99 for validation) and high $R^2$ values of 0.75 (calibration) and 0.80 (validation). When its success was compared against that of the MNLR equation (Equation (13)), it was seen that the empirical equation outperformed the MNLR one, which had produced high errors and very low $R^2$ values of 0.22 (calibration) and 0.17 (validation) (see Table 15). It may be stated that the MNLR method had produced a worse performance for SRE than for WCP. The GA-based empirical model produced comparable results against those of the ANN, as seen in Table 15. However, as it is pointed out above, ANNs do not accomplish any mathematical relations between the dependent and independent variables. They are, as it is pointed out in the literature, very powerful interpolators but poor extrapolators and they are black box models [25]. The empirical models, on the other hand, can be easily employed for both the interpolation and extrapolation purposes.

**Table 15.** Performance comparison between GA_based empirical equation, MNLR, and ANN for SRE.

| SRE | Calibration Stage | | | Validation Stage | | |
|---|---|---|---|---|---|---|
| | **GA** | **MNLR** | **ANN** | **GA** | **MNLR** | **ANN** |
| MAE | 10.32 | 18.04 | 1.47 | 10.77 | 20.14 | 4.32 |
| RMSE | 13.18 | 24.00 | 2.37 | 12.99 | 25.98 | 5.37 |
| $R^2$ | 0.75 | 0.22 | 0.99 | 0.80 | 0.17 | 0.96 |

The above model validation performance results were also summarized by using the Taylor Diagram, which quantifies the degree of correspondence between predicted and measured data [26]. For this purpose, it employs three different statistics called (1) the Pearson correlation coefficient, (2) the centered RMSE, and (3) the Standard deviation. Figures 14 and 15 present these statistics for WCP and SRE performance results for the validation stages, respectively. In these figures, the correlation coefficient is related to the azimuthal angle and shown by black contours; the centered RMSE is represented by green arc contours and blue-dashed contours show the standard deviation. As seen in Figures 14 and 15, the corresponding three statistics for each case are compatible with the performance measures presented in Tables 14 and 15.

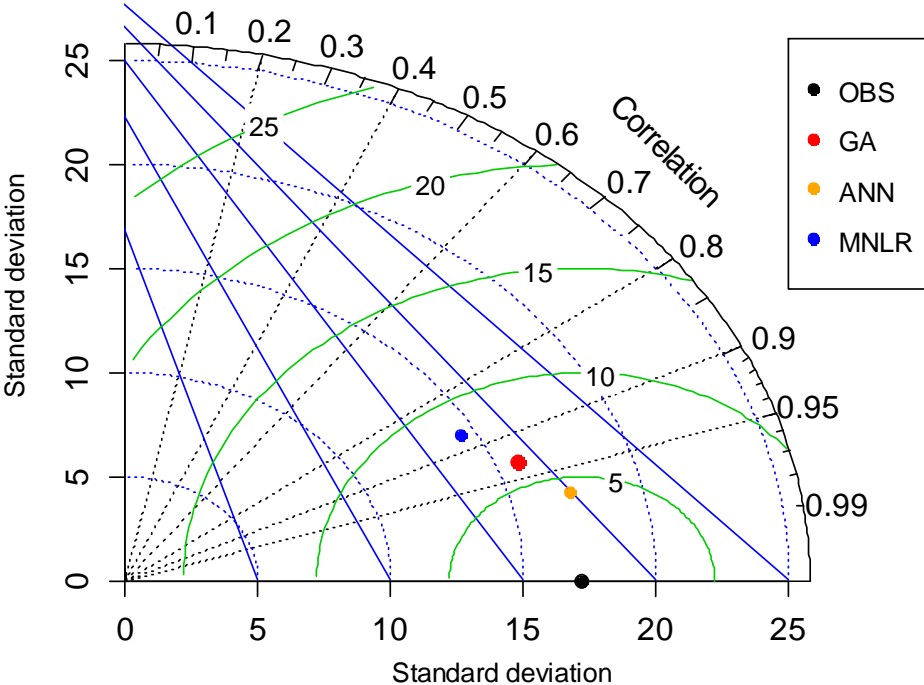

**Figure 14.** Summary of statistics of predicted WCP values by Taylor Diagram (black contours: Pearson correlation coefficient; green contours: centered RMSE; blue contours: standard deviation).

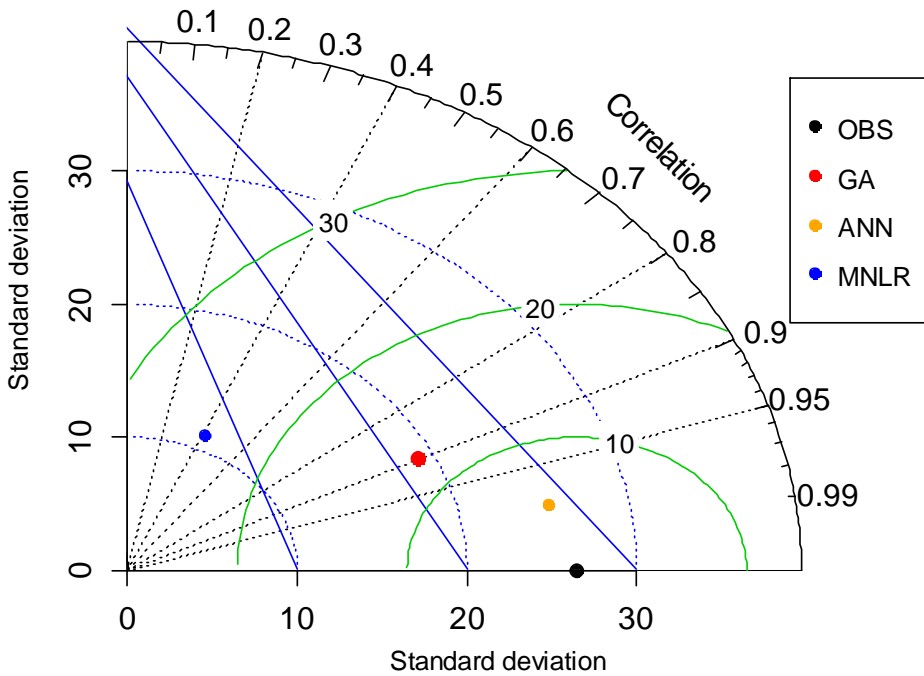

**Figure 15.** Summary of statistics of predicted SRE values by Taylor Diagram (black contours: Pearson correlation coefficient; green contours: centered RMSE; blue contours: standard deviation).

It is worth noting that this study is the first one developing empirical equations for the SRE and the WCP as functions of water, sediment, fluid, and flow parameters and Coanda intake physical characteristics. Based on the results of the experiments, an optimum design based on curvature radius was developed to prevent clogging during the intake operations. As pointed out in the Introduction section, there is a limited number of studies on Coanda intakes, especially subjected to the sediment-laden flows. These studies have involved mostly experimental works that lack any equation development. Therefore, discussing the developed empirical equations within the framework of existing literature becomes quite limited. Hazar and Elci [21] have only attempted to propose multilinear regression (MLR) equations for the variables of SRE and WCP. Apart from the fact that the process is nonlinear rather than linear, they have employed all the data for the calibration stage and they have not verified their equations. In this study, however, the nonlinear empirical equations were calibrated with about 70% of the data and validated using the rest (about 30%). These equations were compared against those of the MNLR ones. The results have shown the superiority of the empirical equations.

Note that when Equation (8) was developed, about 76 sets of data were used in its calibration. The so-developed equation was then employed to make the predictions of the other 32 WCP values, which were not presented to the model at all during its calibration stage. As discussed above, the equation made good predictions of the measured 32 WCP values and outperformed the MNLR model (see Table 14 and Figure 14). This success points to the right form of the equation. In a similar fashion, in the development of Equation (9), 76 sets of data were used in its construction and the rest (32 sets) were employed to test its performance. As discussed above, the equation made successful predictions of SRE values and outperformed the MNLR model (see Table 15 and Figure 15). This also conforms to the right form of the constructed equation.

## 5. Summary and Conclusions

The main aim of this study is to develop nonlinear empirical equations for WCP and SRE, as functions of fluid, sediment, flow, and intake structure parameters for Coanda type intakes. To develop comprehensive but at the same time practical and user-friendly equations, the number of parameters and variables was first reduced from 14 to 10 by creating the dimensionless parameters. Then, the multicollinearity analysis was performed to remove any possible collinearity problem. As a result, only five dimensionless parameters for the WCP and five for the SRE were considered in the development of the equations. Next, the forms of the equations were decided upon by investigating direct or inverse variation between each parameter and the respected output variable of SRE and WCP. The optimal values of the exponents and coefficients of the proposed equations were obtained using the GA. The proposed equations were successfully calibrated with 70% of the data and validated with the rest. The developed equations were then compared against the MNLR and ANN. Results have shown that the GA-based empirical equations have reliable predictive capability and outperform the MNLR ones.

The performance of the GA_based empirical equations is comparable to that of the ANNs that produce less error and high $R^2$ values. Yet, the ANN cannot accomplish neither any mathematical relations nor can they be used for the extrapolation purpose, unlike the empirical equations. The sensitivity analysis results carried out by the ANNs reveal that the geometric characteristics parameters of Coanda intakes are comparably more sensitive than the flow ones for the WCP. Similarly, both the geometric and sediment parameters are more sensitive than the flow characteristics in the case of the SRE.

Note that each nonlinear empirical equation (Equations (8) and (9)) was developed using 76 sets of data and verified by being applied to another 32 validation data sets that were not presented to the model at all during the respected calibration stage. The successful predictions of the validation data conform to the right form of each proposed equation. Therefore, this study concludes that the developed empirical equations can be employed to predict WCP and SRE for Coanda type intakes. It needs to be pointed out, however, that the equations developed in this study have used laboratory data. Thus, it would be suggested to test these equations in field conditions.

With the advantage of reducing the number of variables that describe a system, the empirical equations derived from non-dimensional numbers reduce the number of experiments enabling correlations of physical phenomena to scalable systems. It is noted that all the experiments were conducted using constant angled T-shape bars and a screen length of 60 cm. In a future study, at higher discharge rates and varying angled T-shape bars, different screen lengths can be investigated, and accordingly, the proposed empirical equations can be revised.

**Author Contributions:** Conceptualization, G.T. and O.H.; methodology, G.T. and O.H.; software, O.H.; validation, G.T. and O.H.; formal analysis, G.T.; investigation, O.H.; resources, S.E.; data curation, S.E. and O.H.; writing—original draft preparation, O.H.; writing—review and editing, G.T. and S.E.; supervision, V.P.S. All authors have read and agreed to the published version of the manuscript.

**Funding:** This research received no external funding.

**Institutional Review Board Statement:** Not applicable.

**Informed Consent Statement:** Not applicable.

**Data Availability Statement:** Not applicable.

**Conflicts of Interest:** The authors declare no conflict of interest.

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
