# Peer review of "Developing Predictive Equations for Water Capturing Performance and Sediment Release Efficiency for Coanda Intakes Using Artificial Intelligence Methods"

_water, doi:10.3390/w14060972_

Round 1

Reviewer 1 Report

The main objective of this study is to develop nonlinear empirical equations for Water Capturing Performance and Sediment Release Efficiency, as functions of fluid, sediment, flow, and intake structure parameters for Coanda type intakes. To develop comprehensive but at the same time practical and user-friendly equations, the number of parameters and variables are first reduced from 14 to 10 by creating the dimensionless parameters. Then, the multicollinearity analysis is performed to remove any possible collinearity problem.

The research design is appropriate. The methods and results are adequately described. The results are clearly presented. The conclusions are supported by the results.

This research has important value. However, before publication, there are still some areas in the article that need to be improved. Below is my comment.

Minor spell check required.

Please consider reviewing the abstract. Editors recommend that the abstract should be a total of about 200 words maximum. The abstract should be a single paragraph and should follow the style of structured abstracts but without headings: 1) Background: Place the question addressed in a broad context and highlight the purpose of the study; 2) Methods: Describe briefly the main methods or treatments applied. Include any relevant preregistration numbers, and species and strains of any animals used. 3) Results: Summarize the article's main findings; and 4) Conclusion: Indicate the main conclusions or interpretations. The abstract should be an objective representation of the article: it must not contain results which are not presented and substantiated in the main text and should not exaggerate the main conclusions.

The Introduction should be expanded and reflect current research on the topic of the article. Of the 22 references, only 1 are dated after 2018.
Reference "Hazar and Elci (2021)" is missing from the reference list, there is only "Hazar and Elci (2020)".

In the text, reference numbers should be placed in square brackets [ ], and placed before the punctuation; for example [1], [1–3] or [1,3]. For embedded citations in the text with pagination, use both parentheses and brackets to indicate the reference number and page numbers; for example [5] (p. 10). or [6] (pp. 101–105).

Please consider reviewing the style of References.
Please check this page: https://www.mdpi.com/journal/water/instructions

Please check the text for "Equations (2) and (3)" and "Equations 2 and 3". Use the required format.

Plagiarism was found in the paper.

Author Response

Reviewer #1

Minor spell check required.

REPLY:  It is done so.

Please consider reviewing the abstract. Editors recommend that the abstract should be a total of about 200 words maximum.

REPLY: It is now 195 words.

The abstract should be a single paragraph and should follow the style of structured abstracts but without headings: 1) Background: Place the question addressed in a broad context and highlight the purpose of the study; 2) Methods: Describe briefly the main methods or treatments applied. Include any relevant preregistration numbers, and species and strains of any animals used. 3) Results: Summarize the article's main findings; and 4) Conclusion: Indicate the main conclusions or interpretations. The abstract should be an objective representation of the article: it must not contain results which are not presented and substantiated in the main text and should not exaggerate the main conclusions.

REPLY: It is revised, as suggested.

The Introduction should be expanded and reflect current research on the topic of the article. Of the 22 references, only 1 are dated after 2018.

REPLY: There are not many studes on Coanda intakes. We have re-scanned the literature as much as possible and were able to add few more recent related-citations.

Reference "Hazar and Elci (2021)" is missing from the reference list, there is only "Hazar and Elci (2020)".

REPLY: It is fixed.

In the text, reference numbers should be placed in square brackets [ ], and placed before the punctuation; for example [1], [1–3] or [1,3]. For embedded citations in the text with pagination, use both parentheses and brackets to indicate the reference number and page numbers; for example [5] (p. 10). or [6] (pp. 101–105).

REPLY: It is done so

Please consider reviewing the style of References.
Please check this page: https://www.mdpi.com/journal/water/instructions

REPLY: It is done so

Please check the text for "Equations (2) and (3)" and "Equations 2 and 3". Use the required format.

REPLY: It is done so

Plagiarism was found in the paper.

REPLY: We checked the paper for plagiarism and found it as 7% due to mostly some common definitions of methods. Then, we tried to avoid it as much as possible and were able to reduce it to 6% involving the parameters and their definitions that are inevitable kept in the paper.

Reviewer 2 Report

see attached file

Author Response

Reviewer #2

The focus of the paper is very interesting and few similar studies are available in the literature and it is well known that the development of empirical equation based on experimental data is worthy of investigation. In overall, the manuscript is well written and easy to read and the provided information’s are well organized.  Few amendments can help in better improving the manuscript.

  1. The ANN model is poorly described.

REPLY: We substantially extended the ANN section by providing more information on the mathematical background and network structure.

  1. Provide a figure for the ANN model by highlighting the input and output variables.

REPLY: We added two more ANN figures; one for WCP and one for SRE showing input and output variables for each case.

  1. Figures are for bad quality, please provide better figure (i.e., 5 to 10).

REPLY: It is done so

  1. Provide the boxplot, violin plot and Taylor diagram

REPLY: We have provided Taylor diagrams for WCP and SRE.

Round 2

Reviewer 1 Report

Most of my comments were taken into account and necessary corrections were made. The article looks much better.

Please consider reviewing the abstract again. The abstract is a text description. I would ask you to take out the many numerical results from the abstract. These would have a place in the summary.

Author Response

We have revised the Abstract, as suggested.